# High-Intensity Interval Training is Associated with Improved Long-Term Survival in Heart Failure Patients

**DOI:** 10.3390/jcm8030409

**Published:** 2019-03-25

**Authors:** Chih-Chin Hsu, Tieh-Cheng Fu, Shin-Sheng Yuan, Chao-Hung Wang, Min-Hui Liu, Yu-Chiau Shyu, Wen-Jin Cherng, Jong-Shyan Wang

**Affiliations:** 1Department of Physical Medicine and Rehabilitation, Keelung Chang Gung Memorial Hospital, Keelung 204, Taiwan; steele0618@gmail.com (C.-C.H.); fic6481@gmail.com (T.-C.F.); 2School of Traditional Chinese Medicine, College of Medicine, Chang Gung University, Taoyuan 333, Taiwan; 3Community Medicine Research Center, Keelung Chang Gung Memorial Hospital, Keelung 204, Taiwan; yuchiaushyu@gmail.com; 4Heart Failure Research Center, Cardiology Section, Department of Internal Medicine, Keelung Chang Gung Memorial Hospital, Keelung 204, Taiwan; bearty54@cgmh.org.tw (C.-H.W.); minimin624@gmail.com (M.-H.L.); 5Institute of Statistical Science, Academia Sinica, Taipei 115, Taiwan; 6Division of Cardiology, Department of Internal Medicine, Keelung Chang Gung Memorial Hospital, Keelung 204, Taiwan; 7Department of Nursing, Ching-Kuo Institute of Management and Health, Keelung 203, Taiwan; 8Institute of Molecular Biology, Academia Sinica, Taipei 115, Taiwan; 9Department of Nursing, Chang Gung University of Science and Technology, Taoyuan 333, Taiwan; 10Division of Cardiology, Department of Internal Medicine, Linkou Chang Gung Memorial Hospital, Taoyuan 333, Taiwan; cwenjin@ms3.hinet.net; 11Institute of Rehabilitation Science, Department of Physiotherapy, College of Medicine, Chang Gung University, Taoyuan 333, Taiwan

**Keywords:** heart failure, cardiac rehabilitation, ventricular remodeling, oxygen consumption, cumulative survival rate

## Abstract

This matched-control cohort study explored the effects of high-intensity interval training (HIIT) on left ventricle (LV) dimensions and survival in heart failure (HF) patients between 2009 and 2016. HF patients who underwent the multidisciplinary disease management program (MDP) were enrolled. Non-exercising participants, aged (mean (95% confidence interval)) 62.8 (60.1–65.5) years, were categorized as the MDP group (*n* = 101). Participants aged 61.5 (58.7–64.2) years who had completed 36 sessions of HIIT were treated as the HIIT group (*n* = 101). Peak oxygen consumption (VO_2peak_) and LV geometry were assessed during the 8-year follow-up period. The 5-year all-cause mortality risk factors and overall survival rates were determined in the longitudinal observation. An increased VO_2peak_ of 14–20% was observed in the HIIT group after exercise training. Each 1-mL/kg/min increase in VO_2peak_ conferred a 58% improvement in 5-year mortality. Increased LV end-systolic diameter (LVESD) was significantly (*p* = 0.0198) associated with increased mortality. The 8-month survival rate was significantly improved (*p* = 0.044) in HIIT participants compared to non-exercise participants. HF patients with VO_2peak_ ≥14.0 mL/kg/min and LVESD <44 mm had a significantly better 5-year survival rate (98.2%) than those (57.3%) with lower VO_2peak_ and greater LVESD. Both HIIT-induced increased VO_2peak_ and decreased LVESD are associated with improved survival in HF patients.

## 1. Introduction

Heart failure (HF) is caused by a structural or functional cardiac disorder that impairs ventricular relaxation or ejection [1] and affects approximately 26 million people in the world [2]. Population surveys have estimated HF prevalence at 2–3% in Western societies [3] and 1.26–6.7% in Asia [4]. The 1- and 5-year survival rates after the onset of HF in the Framingham study were approximately 70% and 50%, respectively [5]. The estimated global burden of HF was $108 billion USD in 2012 [6]. Therefore, care for HF patients has become a significant challenge in modern medicine owing to the high mortality and growing medical costs.

Although exercise intensity greater than 80% of peak oxygen consumption (VO_2peak_) corresponds to vigorous physical activity, supervised exercise training beyond this level has been recommended as a safe approach for aging people [7] and HF patients [8,9]. High-intensity interval training (HIIT) has been used as a form of interval training to improve the cardiorespiratory function in recent clinical practice and is characterized by alternating short periods of exercise at ≥80% of one’s VO_2peak_ with less intense (40–50% of VO_2peak_) recovery periods [7]. This exercise strategy has been shown to increase VO_2peak_ by 10–20% in HF patients with different left ventricular ejection fraction (LVEF) [8,9,10]. HIIT also increases red blood cell deformability [11] and promotes cerebral/muscular hemodynamics [9]. Thus, HIIT has been advocated as a beneficial therapy for HF patients [8,9].

Studies have confirmed that moderate-intensity continuous exercise training (MICT) improves LVEF and reduces LV end-diastolic diameter (LVEDD) in patients with LVEF ≤40% (HFrEF: HF with reduced LVEF) [12,13]; these findings could not be detected in HF patients with LVEF >40% (HFpEF: HF with preserved LVEF) [14]. Short-term HIIT-induced anti-remodeling effects on HFrEF patients, including boosted cardiac output (CO) and decreased LV volume, have been reported in previous literature [6,15]. However, the effects of short-term HIIT on overall survival and long-term LV remodeling features in HF patients remain unclear.

The present work was summarized from an 8-year cohort study for progression of LV geometry and survival in HF patients with and without HIIT during the longitudinal follow-up (F/U). With in-depth analysis on these long-term observations, we hoped to provide insights into the HIIT effects on LV remodeling and associated clinical outcomes in different types of HF patients, which might lead to better management for different types of HF patients.

## 2. Methods

### 2.1. Participants

This was a prospective matched-control study on a priori collected data, which followed the REporting of studies Conducted using Observational Routinely collected health Data (RECORD) and the STrengthening the Reporting of OBservational studies in Epidemiology (STROBE) recommendations [16]. The institutional review board of a community hospital approved the study protocol (201601068B0) and the clinical trial registry number is NCT03245125. A total of 541 HF patients diagnosed according to the Framingham heart failure diagnostic criteria [17] were primarily surveyed for the candidate of exercise training between 1 January 2009 and 31 December 2016. Among them, 330 HF patients with stable clinical presentations [18] ≥4 weeks received the multidisciplinary disease management program (MDP) provided by our HF care team, who offered individualized patient education and optimized guideline-based management [19,20]. Those who were ≥80 or <20 years old, were unable to exercise because of a non-cardiac disease, were pregnant, interrupted exercise training during F/U, were lost to F/U at our cardiologic clinic, were candidates for cardiac transplantation within 6 months, were uncompensated HF patients, or were renal patients with an estimated glomerular filtration rate of <30 mL/min/1.73 m^2^ were not eligible for exercise training. Patients who had absolute contraindications for cardiopulmonary exercise test (CPET) and aerobic activities, suggested by the American College of Sports Medicine (ACSM) [21], were also excluded from the study. The 234 HF patients eligible for exercise trainings were further divided into exercise and non-exercise groups based on patient preference. All the participants provided informed consent before initiating the experimental procedure.

### 2.2. Clinical Assessments

Age, sex, and body mass index (BMI) were recorded for all included subjects. Serum levels of low-density lipoprotein (LDL), creatinine (Cre), glycohemoglobin (Hba1c), and cardiac stress-related b-type natriuretic peptide (BNP) [15,22], were scheduled to be checked in all the recruited subjects at baseline and during F/U. The physical component score (PCS) and mental component scores (MCS) in the Medical Outcomes Study Short Form 36 (SF-36) were used to assess the quality of life (QoL) before each CPET. Echocardiography and CPET were performed during long-term F/U. The list of all variables is provided in Appendix A.

#### 2.2.1. Echocardiography

Cardiac images were acquired at end-expiration with a 2–5 MHz tightly curved-array ultrasound transducer (Vivid 7, General Electric Healthcare, Chicago, IL, USA or Phillips IE33, Philips Healthcare, Andover, MA, USA). M-mode echocardiography, proven to be highly reproducible with low variability and wealth of accumulated data [23,24], was used to measure LVEDD, LV end-systolic diameter (LVESD), and LVEF for all subjects. HF patients with LVEF of ≤40% were classified as having HFrEF, and those with LVEF >40% were classified as having HFpEF based on their baseline echocardiographic findings [9].

#### 2.2.2. Cardiopulmonary Exercise Test

The participants underwent an incremental exercise test on a bicycle ergometer (Ergoselect 150P, ergoline GmbH, Bitz, Germany), which was performed at a work-rate of 10 W/min with continuous monitoring heart rate (HR), brachial blood pressure, and arterial oxygen saturation, until the stopping conditions described previously [25]. Oxygen consumption was measured breath by breath using a MasterScreen CPX (CareFusion, Hoechberg, Germany), and the VO_2peak_ was defined as the guideline for exercise testing suggested by the ACSM [21].

### 2.3. Interventions

HF patients with MDP who underwent an additional 36 sessions (2–3 sessions/week) of supervised hospital-based exercise training on a bicycle ergometer in 3–4 months as the previous protocol [9,26] were classified as the HIIT group (MDP+HIIT). The HIIT protocol was performed as five 3-min intervals at 80% VO_2peak_, and each interval was separated by 3-min exercise at 40% VO_2peak_ [9,25]. The exercise training was terminated when the subject had symptoms/signs during HIIT in accordance with the ACSM guideline [21]. Home exercise programs were not suggested for the HIIT participants. The others without supervised exercise training received advice for home-based physical activities and were classified as the MDP group.

### 2.4. Follow-Up

The participants were followed up until 31 December 2016, or when death occurred during the observational period. All the HF patients were assessed at the time of recruitment. All the HIIT participants underwent secondary CPET, echocardiography, and blood tests within 1 week after completing HIIT. The HIIT participants were scheduled to have echocardiographic examinations every 6 months after completing 36 sessions of HIIT. MDP participants were also scheduled to receive the echocardiographic examination every 6 months during F/U. Dates and causes of death were also documented.

### 2.5. Statistical Analysis

#### 2.5.1. Longitudinal Analysis

Baseline demographic information between the HIIT and MDP groups were compared by Student’s *t* test or Mann–Whitney U test (for deceased subjects) for continuous variables, and the chi-squared test for nominal variables. The paired sample *t*-test was used to assess HIIT effects on VO_2peak_, peak HR, peak O_2_ pulse, cardiac geometry, and QoL in both types of HF patients. Data were presented as mean values with 95% confidence intervals (CIs) or as numbers with the percentage.

Propensity score matching (PSM) was used to reduce the bias caused by age, sex, and LVEF in estimating HIIT effects on HF patients [27]. Rebalancing results were described in the Appendix A. To determine the risk and protective factors of 5-year all-cause mortality in the HF patients, the HIIT participants were followed after they had completed 36 HIIT sessions, but the MDP subjects were tracked from inclusion in the study. The choices of these index dates were for the reduction of the immortal time bias. Multivariate Cox regression was conducted to estimate adjusted hazard ratios (aHR) of clinical indicators corresponding to the 5-year all-cause mortality.

The Kaplan–Meier survival curves for HF patients based on their exercise status were assessed by log rank tests. In order to determine the HIIT effects on the 1-year and later long-term survival, landmark analysis was performed with the landmark time set at one year after the initiation of F/U. As blunted CO associated with heightened ventilatory responses during exercise has been observed in HF patients with VO_2peak_ of <14 mL/kg/min [22], the patients were further stratified by their VO_2peak_ and LVESD values for comparing their survival curves. The cut point value for VO_2peak_ was 14 mL/kg/min and that for LVESD was 44 mm, which corresponds to 99.9% of the healthy individual [28]. A *p*-value less than 0.05 was considered as being of statistical significance. 

#### 2.5.2. Functional Principal Component Analysis (FPCA)

Longitudinal changes in cardiac geometry and VO_2peak_ during F/U were used to reflect physiological adaptations to interventions. They were defined as the following:Y (LVEF_Diff) = LVEF_F/U_ − LVEF_baseline_.Y (Normalized LVEDD_Diff) = (LVEDD_F/U_ − LVEDD_baseline_)/LVEDD_baseline_.Y (Normalized LVESD_Diff) = (LVESD_F/U_ − LVESD_baseline_)/LVESD_baseline_.Y (Normalized VO_2peak__Diff) = (VO_2peak_F/U_ − VO_2peak_baseline_)/VO_2peak_baseline_.

In the FPCA data analysis, the longitudinal response Y in each individual was modeled by
$$Y_{i}\left(d\right)=m\left(d\right)+\sum Z_{j}\left(d\right)\beta_{\mathrm{ij}}$$
where Z_j_ was the eigenfunction. The number of eigenfunctions was selected until 95% of variances were explained by the model (Appendix A).

For each subject, the FPCA score (loading coefficients on the space spanned by eigenfunctions) was calculated and the estimated trend for each group was assessed on the basis of the linear combinations of $\sum Z_{j}\left(d\right)\beta_{\mathrm{ij}}$. Given the two trend curves, we used the maximum distance between these two curves, and the permutation tests were used to provide the *p*-value to indicate how unlikely the observed separations of these two curves based on random assignments of the group [29].

## 3. Results

### 3.1. Patient Characteristics

After 1:1 matching via PSM was performed, a total of 202 subjects were included in the analysis, but 32 MDP participants were excluded. They were further divided into the HIIT (72 HFrEF and 29 HFpEF) and MDP (70 HFrEF and 31 HFpEF) groups. The classification, inclusion/exclusion criteria, and F/U algorithm are shown in Figure 1. All HIIT participants exercised as the prescribed intensity. Clinical presentations were similar between the HIIT and MDP participants, except an increase of resting HR in MDP participants (Table 1). Baseline blood chemistries between the two groups were also similar (Appendix A). 

### 3.2. HIIT Improved VO_2peak_ and Quality of Life

All participants had an initial CPET. F/U CPET was performed on all HIIT participants. In the HIIT participants, significant increases (*p* < 0.001) of VO_2peak_ in HFrEF (post-HIIT vs. pre-HIIT = 20.5 mL/kg/min vs. 17.2 mL/kg/min) and HFpEF (post-HIIT vs. pre-HIIT = 18.5 mL/kg/min vs. 16.2 mL/kg/min) patients were observed after completing 36 sessions of exercise training. Peak HR, O_2_ pulse, and PCS also showed this trend. The HFrEF patients but not the HFpEF patients felt better soon after completing HIIT (Table 2).

### 3.3. Increased VO_2peak_ and Decreased LVESD Were Protective Factors Against Mortality

Our results showed that each 1-mL/kg/min increase of VO_2peak_ conferred a significant reduction (*p* = 0.0002) of the 5-year all-cause mortality by approximately 58% (aHR = 0.4224) in HF patients. On the other hand, greater LVESD significantly increased (aHR = 1.0767, *p* = 0.0198) mortality in HF patients (Figure 2).

A significantly better 8-month overall survival rate (*p* = 0.044) was observed in the HIIT participants compared to the MDP participants (Figure 3A). An increased trend of the cumulative mortality was observed in non-exercise participants at the first and after the fifth F/U year (Appendix A). HF patients with VO_2peak_ ≥14 mL/kg/min (*p* < 0.001) or LVESD <44 mm (*p* = 0.017) had significantly better 5-year survival rates than those with different manifestations (Figure 3B,C). Those who had both VO_2peak_ ≥14 mL/kg/min and LVESD <44 mm had the best 5-year overall survival rate of 98.2%, which was significantly better than the rate of 57.3% in those who simultaneously possessed VO_2peak_ <14 mL/kg/min and LVESD ≥44 mm (Figure 3D).

HIIT increased the proportion of HFrEF patients to have LVESD <44 mm (Figure 4A) and HFpEF patients to have VO_2peak_ ≥14 mL/kg/min (Figure 4C). In HIIT participants with HFrEF, those with LVESD <44 mm after completing 36 sessions of HIIT (*p* = 0.005) showed a significant association with improved long-term survival rate compared to those with LVESD ≥44 mm (Figure 4B). In HIIT participants with HFpEF, those with VO_2peak_ ≥14 mL/kg/min after completing 36 sessions of HIIT showed a significant association with the improved (*p* = 0.01) survival rates compared to those with VO_2peak_ <14 mL/kg/min (Figure 4D). Causes of death were similar between the exercise and non-exercise groups (Appendix A). 

### 3.4. HIIT Reversed Cardiac Remodeling In HFrEF Patients

The mean LVEDD was decreased significantly (*p* = 0.002) from 63.2 mm to 60.0 mm, and the mean LVESD was decreased significantly (*p* < 0.001) from 54.8 mm to 45.0 mm when we compared before and after completing HIIT in the HFrEF patients. A significant (*p* < 0.001) increase in LVEF from 26.8% to 48.2% was also found in these subjects (Table 2). The relief of cardiac stress in HFrEF patients after HIIT reflected in the significant decrease (*p* < 0.001) in BNP level (Appendix A). 

Among HIIT participants, the median echocardiographic examination time was 4 (range 2–7), and 24 of them did not receive F/U echocardiographic examinations after completing HIIT. The mean echocardiographic examination interval during exercise training was 4.2 (3.6–4.6) months, which extended to 14.4 (13.4–15.4) months after stopping exercise training. Among MDP participants, the median echocardiographic examination time was 3 (range 1–8), and the mean examination interval was 14.6 (13.8–15.4) months.

During the long-term observation, 35 HFrEF patients had LVESD <44 mm after HIIT, while the other 35 HFrEF patients showed otherwise. Among those patients with reduced LVESD, 24 such patients had F/U LVESD measurements and 18 of them maintained LVESD <44 mm at their most recent measurement. On the other hand, among those patients whose LVESD did not improve to the desired level, 23 of them had subsequent LVESD measurement during F/U. Among the 23 subjects, 3 had reduced LVESD to less than 44 mm, and 20 of them this value remained greater than 44 mm. The 5-year trend curves of echocardiographic findings between the HIIT and MDP participants are shown in Figure 5. Short-term HIIT was significantly associated with reduced LVESD (*p* = 0.0052) in HFrEF patients compared to the MDP participants.

### 3.5. HIIT Induced Mild LV Dilatation and Decreased LVEF in HFpEF Patients

Non-significant increases in LV dimensions and LVEF were detected in the HFpEF patients during HIIT (Table 2). In the 5-year trend curves for them, HIIT non-significantly altered LVESD and LVEDD, but significantly decreased (*p* = 0.0136) LVEF as compared to the MDP subject (Figure 5). 

## 4. Discussion

LV remodeling in HF patients is characterized by chamber enlargement associated with advancing contractile dysfunction regardless of the inciting cause [30]. This investigation is the first to demonstrate that short-term HIIT induces different types of LV remodeling to improve VO_2peak_ in different types of HF patients, which protects against mortality and finally improves survival. Promising anti-LV remodeling effects after HIIT have been identified in HFrEF patients during the 8-year longitudinal follow-up. The HIIT-induced increases of VO_2peak_ in HFpEF patients may be associated with non-significant LV dilatation, and this cardiac response is different from the cardiac adaptation to HIIT in HFrEF patients. Our long-term observations (Figure 3) also support the previous report that the improvement of VO_2peak_ after exercise training is a dominant prognostic marker in the reduction of all-cause mortality [31]. The present work also addressed that HF patients with VO_2peak_ ≥14 mL/kg/min and/or LVESD <44 mm could have better 5-year overall survival rates than the rest of HF patients.

LV volume measurements have long been known as an important surrogate marker for HF patients’ survival [11,12,13]. Reductions in LVEDD (−4 mm to −1 mm) and LVESD (≅−3 mm) accompanied by −7% to 16% changes of LVEF were observed in HFrEF patients with HIIT at 50–95% of VO_2peak_ [8,9,15] or MICT at 60–70% of VO_2peak_ [12,13]. The various alterations of LV dimensions in different exercise training studies for HFrEF patients are considered to be related to the different exercise training styles or prescribed exercise intensities [32,33]. The reported exercise compliance, i.e., how many participants exercised at the prescribed intensities, varied from 49–100% [8,9,15], which may also contribute to the inconsistent conclusions. Significant decreases of LVEDD and LVESD associated with a significant increase of LVEF in HFrEF patients after completing exercise training were identified in the present work. High exercise compliance from all our HIIT participants might enhance the observed anti-LV remodeling effects in this study.

HFpEF patients in the study had mild LV enlargement before exercise training [29], in accordance with a prior report [34]. Non-significant LV dilatation was observed in HFpEF patients during HIIT in this study, which was also identified in the previous report [26]. This phenomenon was similar to patients with LV hypertrophy (LVH) [35]. Besides, LV dilatation is also a typical finding in exercise-induced cardiac remodeling (EICR), which reflects a physiological adaptation to the exercise-related increase of preload in healthy persons [36,37]. Therefore, further investigations are required to determine whether the LV dilatation in HFpEF patients is a downstream presentation of EICR or a cardiac adaptation to LVH.

In comparison with the 4% to 5% increase of VO_2peak_ in HFrEF patients with MICT at 70% heart rate reserve [15,38], our work has shown that HIIT provided a robust stimulus of approximately 20% increase of VO_2peak_ and an associated decrease of LV dimensions in the HFrEF patients. Therefore, the improvement of exercise capacity in HFrEF patients is probably related to the reversal of LV remodeling during HIIT. HIIT increased the VO_2peak_ by approximately 14% in HFpEF patients with non-significant LV enlargement in the study. The improvement of VO_2peak_ in HFpEF patients after HIIT may be majorly caused by peripheral adaptations [26] instead of a central cardiac response because slightly reduced LVEF under normal cardiac pumping function has been detected in HFpEF patients during this long-term F/U.

Outcomes have varied substantially across studies of HF patients with heterogeneity in included participants, classifications of cardiac functions, and care programs [39]. A Canadian hospital-based study for admitted HF patients reported 1-year all-cause mortality of 26% in HFrEF patients and 22% in HF patients with LVEF >50% [40]. A better 2-year all-cause mortality of 15% in HF patients was reported in an Eastern community-based cohort study [19]. A high 1-year survival rate in HF patients without HIIT was also found in the study. Ethnic or genomic differences are proposed to explain the better survivals of East Asian HF patients than those of Western and Central Europe following an acute HF episode [41]. 

Although HIIT was significantly associated with reduced death events only within the first F/U year, an increased trend of cumulative mortality has been observed in non-exercise participants during our long-term F/U. Physiological adaptations after exercise training may protect HIIT participants against death. Suppression of cerebral/muscle hemodynamics and ventilatory abnormality reduces functional capacity in HF patients with VO_2peak_ <14 mL/kg/min [25], which may further induce high 5-year mortality. LVESD ≥44 mm has been demonstrated to be associated with poor long-term prognosis for HF patients, which hints that LV remodeling is closely related to the mortality in cardiac patients [30]. In contrast to the non-significant reduction of all-cause mortality in HF patients under MICT [38], our cohort study has indicated that HIIT might protect HF patients from deaths. The increased maximum exercise capacity [42] in association with the exercise-induced reversal of LV remodeling may be attributable to the improved survival in HF patients.

### Limitations

The European Society of Cardiology has defined HF patients having LVEF of 40–49% as HF with mid-range EF in 2016 [1]. Thus, it is less likely to classify this group of patients without guideline in 2009. All included subjects received several echocardiographic examinations during F/U; however, the majority of subjects do not have the examination near the end of F/U. From the aspect of study design, it is of great benefit to perform the additional examination near the end of F/U. Although propensity score matching was performed to reduce the selection biases, it is our limitation that the residual selection bias might remain in our data. The low number of death events and the choice of M-mode echocardiographic survey for LV geometry also limit our study. Other weakness includes that patients’ physical activities at home were not regularly assessed. Therefore, long-term F/U for the change in lifestyle in our HF patients after exercise training is required.

## 5. Conclusions

This 8-year cohort study has demonstrated that HIIT improves the maximum exercise capacity of HF patients, in accordance with previous investigations [7,8,9,11,15,26], and may become a significant prognostic factor of long-term survival in HF patients. The reversal of pathological LV remodeling in HF patients after exercise training is probably beneficial for their long-term survival. HF patients with VO_2peak_ ≥14 mL/kg/min and/or LVESD <44 mm after HIIT were identified to have a better survival probability in this long-term follow-up.

## Figures and Tables

**Figure 1 jcm-08-00409-f001:**
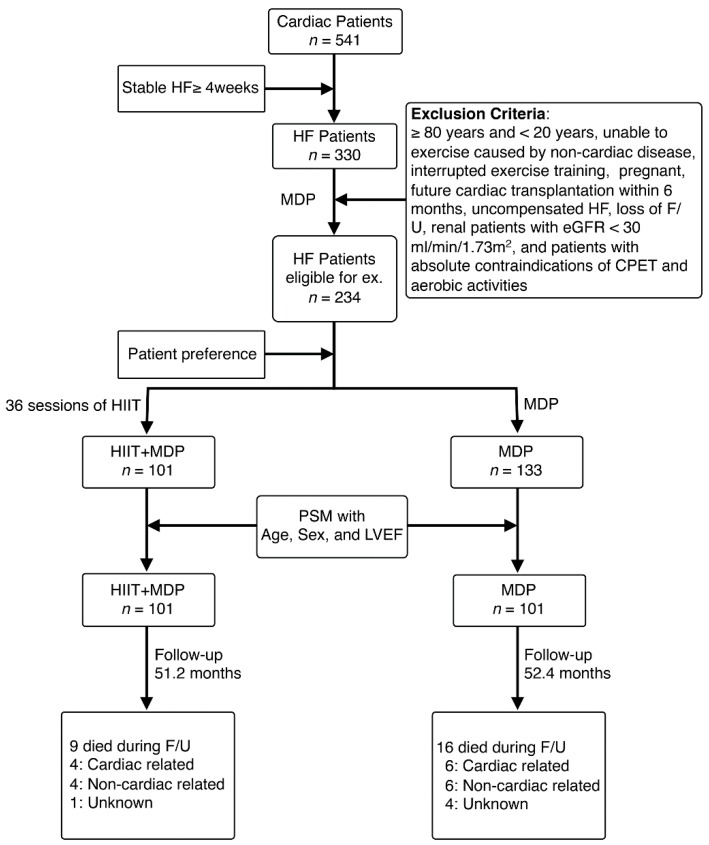
Flowchart of enrolled heart failure (HF) patients during follow-up. The inclusion and exclusion criteria listed in the figure were used to survey HF patients eligible for exercise (ex) training. Enrolled candidates were further divided into simple multidisciplinary disease management program (MDP) or MDP with additional high-intensity interval training (HIIT) groups based on the patient preference. Causes of death were listed at the end of follow-up (F/U). CPET: cardiopulmonary exercise test; eGFR: estimated glomerulus filtration rate; LVEF: left ventricular ejection fraction; PSM: propensity score matching.

**Figure 2 jcm-08-00409-f002:**
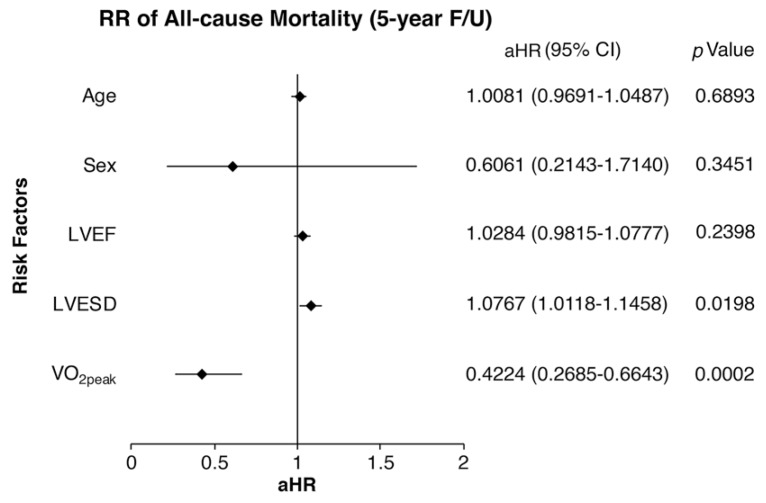
Multivariate Cox regression for estimating the relative risk (RR) of 5-year all-cause mortality in propensity score matching HF patients. Increased LVESD was significantly associated (*p* = 0.0198) with mortality in HF patients. Increased VO_2peak_ significantly reduced the RR of mortality in HF patients. Values of adjusted hazard ratio (aHR) are presented as mean (95% CI).

**Figure 3 jcm-08-00409-f003:**
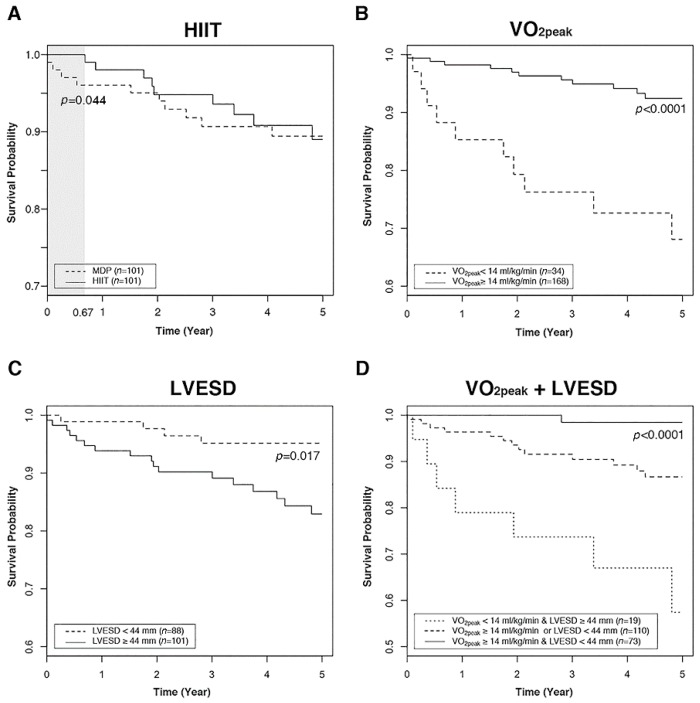
The 5-year overall survival curves analyzed by different categorization in HF patients. (**A**) HIIT participants (—) had a significantly (*p* = 0.044) increased 8-month (gray zone) survival rate compared to MDP participants (---). (**B**) HF patients with VO_2peak_ ≥14 mL/kg/min (—) had significantly better (*p* <0.001) 5-year survivals than those with VO_2peak_ <14 mL/kg/min (---). (**C**) HF patients with LVESD <44 mm (---) had significantly better (*p* = 0.017) 5-year survivals than those with LVESD ≥44 mm (—). (**D**) HF patients with both VO_2peak_ ≥14 mL/kg/min and LVESD <44 mm (—) had significantly better (*p* <0.001) 5-year survivals than those with VO_2peak_ ≥14 mL/kg/min or LVESD <44 mm (---), and those with both VO_2peak_ <14 mL/kg/min and LVESD ≥44 mm (⋯⋯).

**Figure 4 jcm-08-00409-f004:**
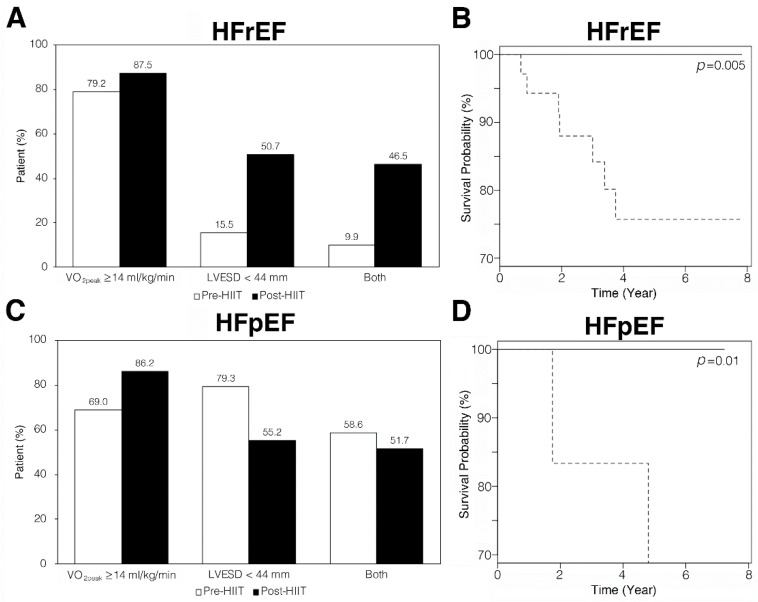
Alterations of VO_2peak_, LVESD, and both of the two indicators in HFrEF and HFpEF patients after HIIT. (**A**) Among HFrEF patients, 15.5% had LVESD <44 mm before exercise training (white bars). Proportion of HFrEF patients with LVESD <44 mm increased to about 50% after HIIT (black bars). (**B**) HFrEF patients with LVESD <44 mm (—) had a significantly greater survival probability (*p* = 0.005) than those with LVESD ≥44 mm (---). (**C**) Among HFpEF patients, 69.0% had VO_2peak_ ≥14 mL/kg/min before exercise training (white bars). The proportion of HFpEF patients with VO_2peak_ ≥14 mL/kg/min increased to 86.2% after HIIT (black bars). (**D**) HFpEF patients with VO_2peak_ ≥14 mL/kg/min (—) had significantly greater survival probability (*p* = 0.01) than those with VO_2peak_ <14 mL/kg/min (---).

**Figure 5 jcm-08-00409-f005:**
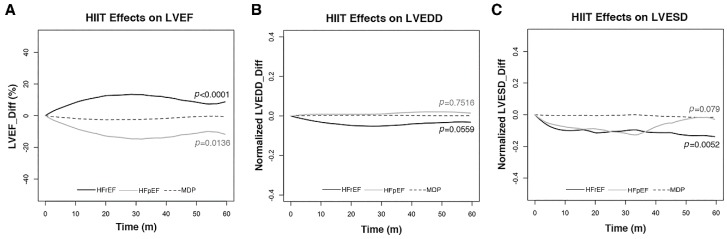
Five-year trend curves for HIIT effects on LV dimensions. Differences of LVEF (LVEF_Diff), LVEDD (Normalized LVEDD_Diff), and LVESD (Normalized LVESD_Diff) in HIIT participants with HFrEF (—), HFpEF (—), or MDP (---) were shown. (**A**) LVEF increased significantly in HFrEF (*p* <0.0001), but decreased significantly in HFpEF (*p* = 0.0136) patients compared to the MDP participants. (**B**) In the HIIT group, LVEDD decreased in HFrEF, but increased in HFpEF patients non-significantly compared to the MDP participants. (**C**) In the HIIT group, LVESD decreased significantly (*p* = 0.0052) in HFrEF patients, but non-significantly in HFpEF patients compared to the MDP participants.

**Table 1 jcm-08-00409-t001:** Baseline demographics of enrolled heart failure (HF) patients.

		HIIT + MDP (*n* = 101)	MDP (*n* = 101)	*p*-Value
HFrEF/HFpEF		72/29	71/30	1.000
Age, years		61.5 (58.7–64.2)	62.8 (60.1–65.5)	0.492
Sex (F/M)		31/70	27/74	0.641
BMI, kg/m^2^		25.7 (24.7–26.6)	25.2 (24.3–26.2)	0.504
HF duration, month		7.76 (4.50–11.0)	11.3 (7.39–15.3)	0.168
NYHA Functional Class, *n* (%)				
I		2 (2)	7 (7)	0.109
II		75 (74)	78 (77)	
III		24 (24)	16 (16)	
Etiology, *n* (%)				
CAD		47 (47)	46 (46)	1.000
DCM		19 (19)	21 (21)	0.860
Hypertension		56 (55)	62 (61)	0.475
Comorbidity, *n* (%)				
Hyperlipidemia		50 (50)	50 (50)	1.000
DM		42 (42)	40 (40)	0.886
Arrhythmia		34 (34)	33 (33)	1.000
Resting BP, mmHg	SBP	123 (119–128)	127 (122–131)	0.318
	DBP	76 (73–79)	77 (74–80)	0.776
Resting HR, bpm		77 (75–79)	81 (78–84)	0.040 ^b^
LVEF, %		34.3 (31.3–37.2)	37.0 (34.1–39.9)	0.190
BNP, pg/mL ^a^		667 (508–801)	635 (482–787)	0.765
Medication, *n* (%)				
ACEI/ARB		82 (81)	83 (82)	1.000
β-blocker		80 (79)	81 (80)	1.000
Diuretics		56 (55)	63 (62)	0.391
MRA		20 (20)	11 (11)	0.117

Baseline information was obtained at recruitment or before the intervention. Values are mean (95% confidence interval, CI) or *n* (%). ACEI: angiotensin-converting-enzyme inhibitor; ARB: angiotensin receptor blocker; BMI: body mass index; BNP: b-type natriuretic peptide; BP: blood pressure; bpm, beats per minute; CAD: coronary artery disease, DBP: diastolic blood pressure; DCM: dilated cardiomyopathy; DM: diabetes mellitus; F/M: female/male; F/U: follow-up; HFrEF: heart failure with left ventricular ejection fraction ≤40%; HFpEF: heart failure with left ventricular ejection fraction >40%; HIIT: high-intensity interval training; HR: heart rate; LVEF: left ventricular ejection fraction; MDP: multidisciplinary disease management program; MRA: mineralocorticoid receptor antagonist; NYHA: New York Heart Association; SBP: systolic blood pressure. ^a^ 79 subjects in exercise and 60 subjects in non-exercise groups received baseline BNP examinations. ^b^ Statistical significance was assessed by Student’s *t*-test.

**Table 2 jcm-08-00409-t002:** Effects of HIIT on exercise capacity, cardiac remodeling, and quality of life.

Type	Assessment		Initial	End-HIIT	*p*-Value
HFrEF	LVEF, %		26.8 (24.6–28.9)	48.2 (44.3–52.1)	<0.001 ^a^
	LVEDD, mm		63.2 (60.7–65.7)	60.0 (57.5–62.5)	0.002 ^a^
	LVESD, mm		54.8 (52.2–57.4)	45.0 (41.9–48.2)	<0.001 ^a^
	VO_2peak_, mL/kg/min		17.2 (16.3–18.1)	20.5 (19.2–21.8)	<0.001 ^a^
	Peak HR, bpm		132 (126–137)	139 (133–146)	0.001 ^a^
	Peak O_2_ pulse, mL/beat		9.22 (8.61–9.82)	10.2 (9.48–10.9)	<0.001 ^a^
	SF-36	PCS	47.0 (45.0–48.9)	50.9 (48.8–52.9)	<0.001 ^a^
		MCS	45.1 (42.8–47.3)	47.6 (45.2–49.9)	0.012 ^a^
HFpEF	LVEF, %		52.9 (48.5–57.2)	53.1 (48.2–57.9)	0.803
	LVEDD, mm		54.3 (51.3–57.4)	56.3 (53.0–59.6)	0.285
	LVESD, mm		38.7 (35.4–42.0)	40.6 (37.1–44.1)	0.387
	VO_2peak_, mL/kg/min		16.2 (15.1–17.4)	18.5 (16.8–20.2)	<0.001 ^a^
	Peak HR, bpm		136 (126–146)	144 (133–154)	0.010 ^a^
	Peak O_2_ pulse, mL/beat		8.49 (7.45–9.52)	9.38 (8.19–10.6)	0.008 ^a^
	SF-36	PCS	45.9 (43.0–48.8)	50.8 (47.6–53.9)	0.005 ^a^
		MCS	42.6 (37.9–47.2)	46.5 (42.3–50.8)	0.084

All participants had baseline CPET with echocardiography examination, and subsequent F/U examinations. Values were mean (95% CI). bpm: beats per minute; HFrEF: heart failure with left ventricular ejection fraction ≤40%; HFpEF: heart failure with left ventricular ejection fraction >40%; LVEDD: left ventricle end-diastolic diameter; LVESD: left ventricle end-systolic diameter; LVEF: left ventricular ejection fraction; MCS: mental component score; PCS: physical component score; SF-36: short form 36; VO_2peak_: peak oxygen consumption. ^a^ Statistical significance was assessed by paired *t*-test.

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
