# Peer review of "High-Intensity Interval Training is Associated with Improved Long-Term Survival in Heart Failure Patients"

_jcm, 2019, doi:10.3390/jcm8030409_

Reviewer 1 Report

Authors present interesting and important study about the benefits of high intensity interval training program in patients with heart failure. In general it is an interesting and well designed study. Its presentation, however, could be improved.

A look at the charts suggests that there is no effect on long term mortality – the statistical effect in Kaplan Meier analysis is in fact observed only in first 2 years. Therefore, in order to be scientificaly sound, authors should perform also a long- term landmark survival analysis and base their conlcusions on middle term (approx one year) and later survival. This confirms previous studies that show dissipation of effects on mortality after 1-2 years

 Authors should precisely define the moment when they analyzed the changes ion echo and CPET parameters – just after HIIT or at the end of the follow-up. And which results are presentd when the improvement s described.

If echo and CPET was not performed at the end of the follow-up authors should add the limitation of the study, that no additional data in terms of long-term effects of HIIT is provided.

Once landmark survival analysis is perfomed and information concerning time of echo and CPET provided,their conclusion should be changed appropriately

Minor remark:

Line 84 Methods  "were also excluded in the study." were those patients excluded from the study or included in?

Author Response

Response to the reviewer 1

1.     A look at the charts suggests that there is no effect on long term mortality – the statistical effect in Kaplan Meier analysis is in fact observed only in first 2 years. Therefore, in order to be scientifically sound, authors should perform also a long-term landmark survival analysis and base their conclusions on middle term (approx one year) and later survival. This confirms previous studies that show dissipation of effects on mortality after 12 years.

Reply: Thank you for your comment. Although the difference of the long-term survival probability between exercise and non-exercise participants was not significant, the increased trend of cumulative mortality was observed in non-exercise participants during the first F/U and after the 5th follow-up year (landmark analysis in the supplemental figure 3). The very small number of deaths may cause statistical insignificance. We agreed that the dissipation effects on mortality are evident after 12 months. We have placed the description in Line 965-967, 2nd paragraph in sub-section 3.3., Page 7/14.

 2.     Authors should precisely define the moment when they analyzed the changes ion echo and CPET parameters – just after HIIT or at the end of the follow-up. And which results are presented when the improvements described. If echo and CPET was not performed at the end of the follow-up authors should add the limitation of the study, that no additional data in terms of long-term effects of HIIT is provided. Once landmark survival analysis is performed and information concerning time of echo and CPET provided, their conclusion should be changed appropriately

Reply: Thank you for your comment. We have re-written the follow-up CPET and echocardiographic examinations in the sub-section 2.4., Page 3/14.

All the HF patients were assessed at the time of recruitment. All the HIIT participants underwent secondary CPET, echocardiography, and blood tests within 1 week after completing HIIT. The HIIT participants were scheduled to have echocardiographic examinations every 6 months after completing 36 sessions of HIIT. MDP participants were also scheduled to receive the echocardiographic examination every 6 months during F/U.

The examination time and interval between two examinations were described in the 2nd paragraph of the sub-section 3.4.

Among HIIT participants, the median echocardiographic examination time was 4 (range 2-7), and 24 of them did not receive F/U echocardiographic examinations after completing HIIT. The mean echocardiographic examination interval during exercise training was 4.2 (3.6-4.6) months, which extended to 14.4 (13.4-15.4) months after stopping exercise training. Among MDP participants, the median echocardiographic examination time was 3 (range 1-8), and the mean examination interval was 14.6 (13.8-15.4) months.

Indeed, the CPET and echocardiographic examination was not performed at the end of the follow-up. However, there were several echocardiographic examinations in each included subject during F/U. We had placed this description in the limitations paragraph (Page 11/14).

Minor remark:

1.     Line 84 Methods "were also excluded in the study." were those patients excluded from the study or included in?

Reply: Thank you for your comment. We have corrected this error as “excluded from the study” in line 192, sub-section 2.1., Methods section, Page 2/14.

Reviewer 2 Report

The manuscript presents an extraordinary and well perfomed matched-control cohort study that provides 8-year real-world evidence on high-intensity interval training effects on heart failure patients survival and their left ventricle remodeling. Nevertheless, the entire manuscript must be revised by a native English speaker, particularly by the chaotic use of verb tenses, and two details on the study design and methods should be provided, as they may not be completely accessible to non-experienced readers, nor to scientists of other research areas:

Please, state in the text (in subseciton 2.1. Participants, section 2. Methods): this is a prospective matched-control study on a priori collected data.

Provide information on adherence to RECORD  and STROBE guidelines (in subseciton 2.1. Participants, section 2. Methods). Guidance on overlapping RECORD items 6, 7 and 12 are in PLoS Med 2015;12:e1001885, and should also be explained in a new supplementary methods section.

Author Response

Response to the Reviewer 2

1.     The manuscript presents an extraordinary and well performed matched control cohort study that provides 8-year real world evidence on high-intensity interval training effects on heart failure patients survival and their left ventricle remodeling. Nevertheless, the entire manuscript must be revised by a native English speaker, particularly by the chaotic use of verb tenses, and two details on the study design and methods should be provided, as they may not be completely accessible to non-experienced readers, nor to scientists of other research areas.

Reply: Thank you for your appreciation. The English in the entire manuscript, the design and methods in particular, has been edited by Vikas Narang of Editage according to the reviewer’s suggestions.

2.     Please, state in the text (in subseciton 2.1. Participants, section 2. Methods): this is a prospective matched control study on a priori collected data. Provide information on adherence to RECORD and STROBE guidelines (in subseciton 2.1. Participants, section 2. Methods). Guidance on overlapping RECORD items 6, 7 and 12 are in -, and should also be explained in a new supplementary methods section.

Reply: Thank you for your comment. We have stated “this is a prospective matched control study on a priori collected data” in sub-section 2.1. Participants, section 2. Methods (Line 179, page 2/14).

Reviewer 3 Report

In this study, Hsu et al. included 202 heart failure patients of whom all received a “multidisciplinary disease management program” (MDP) including individual patient education and optimized guideline-based management. Further, half of the patients underwent an additional high-intensity interval training (HIIT) program including 36 occasions of supervised bicycle exercise training. The patients were initially examined with echocardiography for LVEF assessment by Teichholz method and cardiopulmonary exercise testing. These exams were repeated one week after completing HIIT (for the HIIT patients) and at the same time interval for the non-HIIT patients. Thereafter, the patients were followed regarding all-cause death. Using propensity score on age, sex, and LVEF, HIIT-patients and non-HIIT patients were matched. Results included a significant increase in peakVO2 and QoL in HIIT-patients at follow-up and an increase in LVEF in the patients with LVEF<40% at baseline. The 8 month-survival was significantly better in HIIT patients compared with non-HIIT patients.

The paper gives a messy impression (for instance, where is Table 1??)  and definitively needs English language editing. The research questions are interesting and scientifically sound, but the study design is not at all appropriate to answer these questions. The description on of how patients are allocated to intervention is unclear, but it is obvious that they were not randomly selected. Therefore, my main concern is the very high risk of selection bias. Without an appropriate control group, it is impossible to conclude that the demonstrated differences between pre- and post-intervention are caused by the HIIT.

I have a few specific comments below.

The Teichholz method for assessment of LVEF is outdated and advised against by guidelines (Lang et al. J Am Soc Echocardiogr 2005;18:1440).

The allocation of patients to HIIT must be clarified.

Detailed description of baseline characteristics is missing (maybe because Table 1 is missing?). Information on age, sex, body mass index, prevalence of hypertension, diabetes, ischemic heart disease, laboratory data (creatinine, Hb, NT-proBNP), medication etc. is of utmost importance.

The survival analysis was based on a very small number of deaths (n=25) with high risk of low power and overfitting. The 8 month-survival was significantly better in HIIT patients compared with non-HIIT patients. But why testing at 8 months? Seems like a post-hoc choice to me.

First sentence in Results (line 158) is better placed under Statistics.

Redundant information regarding methodological issues already mentioned in Methods is repeated in the Results section (e.g. sentence starting at line 176, two sentences starting at line 194 etc).

The sentence starting at line 253 is better placed under Discussion.

Figure 5 is hard to grasp because of a messy legend. For instance, the line types must be more clearly explained.

Author Response

Response to the Reviewer 3

1.     The paper gives a messy impression (for instance, where is Table 1??) and definitively needs English language editing. The research questions are interesting and scientifically sound, but the study design is not at all appropriate to answer these questions. The description on of how patients are allocated to intervention is unclear, but it is obvious that they were not randomly selected. Therefore, my main concern is the very high risk of selection bias. Without an appropriate control group, it is impossible to conclude that the demonstrated differences between pre- and post-intervention are caused by the HIIT.

Reply: Thank you for your comment. We don’t know why the Table 1 is missing in our submitted manuscript. We attached the Table 1 on the page 6/14 (before sub-section 3.2. HIIT improved VO2peak and quality of life, section 3. Results). In our Table 1, the baseline demography between exercise and non-exercise groups were non-significant, except the resting heart rate. However, this statistical significance can be neglected clinically because subjects in both groups had physiologically normal resting heart rate. In addition, the patient allocation has been clarified in the 2.1. participants (Line 179-193, Page 2/14) and the revised Figure 1 (Page 5/14).

2.     The Teichholz method for assessment of LVEF is outdated and advised against by guidelines (Lang et al. J Am Soc Echocardiogr 2005;18:1440).

Reply: Thank you for your comment. Indeed, M-mode echocardiography was not recommended in current guide line. However, this method is highly reproducible with low intra-/inter-observer variability, and had wealth of accumulated data. Therefore, it has been recommended to screen a large population. We have addressed the reason why we’ve used this technology to measured LV geometry in Line 437-440, sub-section 2.2.1., Page 3/14. Recommended reference has been placed in the revised reference 21.

3.     The allocation of patients to HIIT must be clarified.

Reply: Thank you for your comment. The patient allocation has been clarified in the 2.1. participants (Line 179-193, Page 2/14) and the revised Figure 1 (Page 5/14).

4.     Detailed description of baseline characteristics is missing (maybe because Table 1 is missing?). Information on age, sex, body mass index, prevalence of hypertension, diabetes, ischemic heart disease, laboratory data (creatinine, Hb, NTproBNP), medication etc. is of utmost importance.

Reply: Thank you for your comment. We don’t know why the Table 1 is missing in our submitted manuscript. We attached the Table 1 on the page 6/14 (before sub-section 3.2. HIIT improved VO2peak and quality of life, section 3. Results). In our Table 1, the baseline demography between exercise and non-exercise groups were non-significant, except the resting heart rate. However, this statistical significance can be neglected clinically because subjects in both groups had physiologically normal resting heart rate. In addition, the patient allocation has been clarified in the 2.1. participants (Line 179-193, Page 2/14) and the revised Figure 1 (Page 5/14). Part of the laboratory data was listed in the supplemental Table 1 to reduce the Table 1 size.

5.     The survival analysis was based on a very small number of deaths (n=25) with high risk of low power and overfitting. The 8-month survival was significantly better in HIIT patients compared with non-HIIT patients. But why testing at 8 months? Seems like a post-hoc choice to me.

Reply: Thank you for your comment. Although the difference of the long-term survival probability between exercise and non-exercise participants was not significant, the increased trend of cumulative mortality was observed in non-exercise participants during the first F/U and after the 5th follow-up year (landmark analysis in the supplemental figure 3). This observation may be caused by the very small number of deaths. The low death rate may also result in the significant survival difference only in the short observation period in this longitudinal observation study. We have placed the description in Line 965-967, 2nd paragraph in sub-section 3.3., Page 7/14.

6.     First sentence (Data were presented as mean values with 95% confidence intervals (CIs) or as numbers with the percentage.) in Results (line 158) is better placed under Statistics.

Reply: Thank you for your comment. We have placed the first sentence in the Results section in the statistical analysis sub-section (Line 472-473, sub-section 2.5.1, Methods section, Page 3/14).

7.     Redundant information regarding methodological issues already mentioned in Methods is repeated in the Results section (e.g. sentence starting at line 176, two sentences starting at line 194 etc).

Reply: The cardiopulmonary exercise tests and echocardiographic examinations in the results section were redundant and have been deleted in the revised manuscript.

8.     The sentence starting at line 253 is better placed under Discussion

Reply: The sentence “It is assumed that this trend of LV remodeling maintained during the long-term F/U. Therefore, it is worth sequential assessing LV dimensions in HF patients during long-term F/U, because this measurement accurately reflected LV remodeling status” is not adequate in the Result section, and has been deleted in our revised manuscript.

9.     Figure 5 is hard to grasp because of a messy legend. For instance, the line types must be more clearly explained.

Reply: Thank you for your comment. We have modified our Figure 5 in our revised manuscript. The original 4-panel figure has been changed to a 3-panel figure. We’ve also displayed different line types in the revised figure legend (Page 10/14).

Reviewer 4 Report

Abstract

Aged about 60 years too vague, please state mean/SD of each group. This is especially important as it a matched-control cohort.

LVEF<40% is="" heart="" failure="" with="" reduced="" ejection="" fraction="" lvef="">40% is HFpEF please name these sub-groups as suggested.

Main Text

What does ‘repeatedly’ mean? Every week/month/year? 3-6 month assessments are rather vague timings and unsystematic.

Please use past tense eg ‘hope’ should be ‘hoped’

80% peak VO2 corresponds to vigorous intensity not high intensity according to established guidelines Norton, K. JSAMS 2011.

Please justify how the study can be adequately powered to detect changes between HFpEF groups with only 29 vs 31 participants?

Why were 1/3 of MDP patients lost and none in the vigorous intensity group?

36 times of training is grammatically incorrect. 36 sessions would be better, how often were these sessions per week? This all seems very vague.

The authors make great assumptions that 12 weeks of exercise can reduce mortality up to 5 years later….can evidence be provided to support this notion?

The LVEF% data in HFrEF group are astounding. So most patients went from HFrEF to HFpEF, with a mean 21.4% improvement in LVEF……..Wisloff’s 2007 study (which did use HIIT not vigorous exercise) is considered somewhat of an ‘outlier’ as the results were so good, but they only achieved a 6% improvement in LVEF. Can the authors comment in the discussion on the accuracy of the LEVF measures and why their results were 3x greater than the best previous data which is somewhat of an outlier, while only using a lower (vigorous) intensity exercise?

Author Response

Response to the Reviewer 4

1.     Aged about 60 years too vague, please state mean/SD of each group. This is especially important as it a matched control cohort.

Reply: Thank you for your comment. We have placed the age (mean (95% CI)) in the abstract section (Line 28-30, Page 1/14) according to the reviewer’s suggestion.

2.     LVEF<40% is="" heart="" failure="" with="" reduced="" ejection="" fraction="" lvef="">40% is HFpEF please name these subgroups as suggested.

Reply: Thank you for your comment. We have renamed our HFlow and HFhigh to HFrEF and HFpeF, respectively.

Main Text

1.     What does ‘repeatedly’ mean? Every week/month/year? 36 month assessments are rather vague timings and unsystematic.

Reply: The word “repeatedly” in the abstract section has been deleted. The detail for the examination frequency has been described in the 2.4. and the results were shown in the Line 1164-1169, 2nd paragraph of the sub-section 3.4.

2.4.All the HF patients were assessed at the time of recruitment. All the HIIT participants underwent secondary CPET, echocardiography, and blood tests within 1 week after completing HIIT. The HIIT participants were scheduled to have echocardiographic examinations every 6 months after completing 36 sessions of HIIT. MDP participants were also scheduled to receive the echocardiographic examination every 6 months during F/U.

3.4. Among HIIT participants, the median echocardiographic examination time was 4 (range 2-7), and 24 of them did not receive F/U echocardiographic examinations after completing HIIT. The mean echocardiographic examination interval during exercise training was 4.2 (3.6-4.6) months, which extended to 14.4 (13.4-15.4) months after stopping exercise training. Among MDP participants, the median echocardiographic examination time was 3 (range 1-8), and the mean examination interval was 14.6 (13.8-15.4) months.

2.     Please use past tense eg ‘hope’ should be ‘hoped’.

Reply: Thank you for your comment. We have changed to “hoped” in the last paragraph of the introduction section (Line 170). We also used past tense in our Methods section.

3.     80% peak VO2 corresponds to vigorous intensity not high intensity according to established guidelines Norton, K. JSAMS 2011.

Reply: Thank you for your comment. We have addressed this concept in the second paragraph of the Introduction (Page 2/14) in the revised manuscript.

Although exercise intensity greater than 80% of peak oxygen consumption (VO2peak) corresponds to vigorous physical activity, supervised exercise training beyond this level has been recommended as a safe approach for aging people and HF patients. High-intensity interval training (HIIT) has been used as a form of interval training to improve the cardiorespiratory function in recent clinical practice and is characterized by alternating short periods of exercise at  ³80% of one’s VO2peak with less intense (40-50% of VO2peak) recovery periods. This exercise strategy has shown to increase VO2peak by 10-20% in HF patients with different left ventricular ejection fraction (LVEF).

4.     Please justify how the study can be adequately powered to detect changes between HFpEF groups with only 29 vs 31 participants?

Reply: Based on our earlier study, the effect of HIIT on VO2peak in HFpEF patients was about 20% increase over baseline. The average pre-HIIT VO2peak was 16.22 ml/kg/min. The alternative mean was 19.46 ml/kg/min. Using two-sided test and 4.30 as our standard deviation. We need 28 patients to have at least 80% power. Therefore, the HFpEF patient number in the study was adequate for analysis.

5.     Why were 1/3 of MDP patients lost and none in the vigorous intensity group?

Reply: Thank you for your comment. “After propensity score matching for correcting selection bias, 32 MDP participants were excluded, and totally 202 subjects were included in the analysis” has been described in the first sentence of the sub-section 3.1. (Line 529-630).

6.     36 times of training is grammatically incorrect. 36 sessions would be better. how often were these sessions per week? This all seems very vague.

Reply: Thank you for your comment. We have corrected this error according to the reviewer’ comment. The exercise training protocol has been described in the first sentence (Line 451-452) in the sub-section 2.3., Page 3/14.

7.     The authors make great assumptions that 12 weeks of exercise can reduce mortality up to 5 years later….can evidence be provided to support this notion?

Reply: Thank you for your comment. It is considered that the improvement of the long-term clinical outcome after exercise training may be due to the change of lifestyle after exercise training. However, we did not assess the physical activity at home in the study. Therefore, we have addressed this disadvantage in the Limitations (Line 1726-1728, Page 11/14).

8.     The LVEF% data in HFrEF group are astounding. So most patients went from HFrEF to HFpEF, with a mean 21.4% improvement in LVEF……..Wisloff’s 2007 study (which did use HIIT not vigorous exercise) is considered somewhat of an ‘outlier’ as the results were so good, but they only achieved a 6% improvement in LVEF. Can the authors comment in the discussion on the accuracy of the LEVF measures and why their results were 3x greater than the best previous data which is somewhat of an outlier, while only using a lower (vigorous) intensity exercise?

Reply: M-mode echocardiography, proven to be highly reproducible with low intra-/inter-observer variability and wealth of accumulated data, has been recommended to screen a large population (Line 437-440, sub-section 2.2.1., Page 3/14). Therefore, the measurement of LV dimensions is reliable in the study. The various alterations of LV dimensions in different exercise training studies for HFrEF patients are considered to be related to the different exercise training styles or prescribed exercise intensities. The exercise compliance, i.e., how many participants exercised at the prescribed intensities, which may also contribute to the inconsistent conclusions (the 2nd paragraph in the Discussion section).

Round  2

Reviewer 2 Report

The authors should be able to demonstrate that the study adhere to both the REporting of studies Conducted using Observational Routinely collected health Data (RECORD) and the STrengthening the Reporting of OBservational studies in Epidemiology (STROBE) recommendations in order to adequately judge the validity and significance of findings. RECORD items 6 (selection of study participants), 7 (potential confounders) and 12 (data access) overlapping key STROBE items should be specifically addressed.

1. As this concern from my first report has not been addressed, I suggest to state in subsection Participants, section Methods:

"This is a prospective ... data, that follows the REporting of studies Conducted using Observational Routinely collected health Data (RECORD) and the STrengthening the Reporting of OBservational studies in Epidemiology (STROBE) recommendations. (add this reference at the end of this sentence: Benchimol EI, Smeeth L, Guttmann A, et al. The REporting of studies Conducted using Observational Routinely-collected health Data (RECORD) statement. PLoS Med 2015;12:e1001885. http://dx.doi.org/10.1371/journal.pmed.1001885)"

"541 cardiac patients... January 1, 2009 (RECORD item 6)."

"Among them, 330 HF... guideline-based management (RECORD item 7)"

2. State also in subsection Clinical assessments, section Methods:

"Age, sex,... and during F/U (RECORD item 12).

Author Response

The authors should be able to demonstrate that the study adhere to both the REporting of studies Conducted using Observational Routinely collected health Data (RECORD) and the STrengthening the Reporting of OBservational studies in Epidemiology (STROBE) recommendations in order to adequately judge the validity and significance of findings. RECORD items 6 (selection of study participants), 7 (potential confounders) and 12 (data access) overlapping key STROBE items should be specifically addressed.

1.     As this concern from my first report has not been addressed, I suggest to state in subsection Participants, section Methods: "This is a prospective ... data, that follows the REporting of studies Conducted using Observational Routinely collected health Data (RECORD) and the STrengthening the Reporting of OBservational studies in Epidemiology (STROBE) recommendations. (add this reference at the end of this sentence: Benchimol EI, Smeeth L, Guttmann A, et al. The REporting of studies Conducted using Observational Routinelycollected health Data (RECORD) statement. PLoS Med 2015;12:e1001885. http://dx.doi.org/10.1371/journal.pmed.1001885)"

Reply: We apologize for misunderstanding your previous comments. We have added the above statement in Line 179-182, sub-section 2.1., Participants, Methods section, Page 2/14, and the article in the revised reference 16. We also went over our manuscript and found that there are indeed some descriptions without clear definitions. We also followed the RECORD statement and modify the manuscript.

2.     "541 cardiac patients... January 1, 2009 (RECORD item 6)."; "Among them, 330 HF... guideline based management (RECORD item 7)"

Reply: Thank you for your comment. The detailed criteria for the inclusion of 541 patients and how we reduced them to 330 are updated. We have reorganized and clarified the sub-section 2.1. The screening criteria has also been listed in the revised reference 17-21. We believed that the readers now can repeat exactly how we did it by the information provided in the manuscript.

(541 HF patients diagnosed according to the Framingham heart failure diagnostic criteria [McKee et al. N. Engl. J. Med. 1971, 285, 1441-1446] were primarily surveyed for the candidate of exercise training between January 1, 2009 and December 31, 2016.  Among them, 330 HF patients with stable clinical presentations [Working Group on Cardiac Rehabilitation & Exercise Physiology; Working Group on Heart Failure of the European Society of Cardiology. Eur. Heart J. 2001, 22,37-45] 4 weeks received the multidisciplinary disease management program (MDP) provided by our HF care team, who offered individualized patient education and optimized guideline-based management [Lee et al. Int. Heart J. 2012, 53, 364-369; Mao et al. J. Cardiovasc. Med. (Hagerstown) 2015, 16, 616-624]. Those who were ≥ 80 or < 20 years old, unable to exercise because of a non-cardiac disease, pregnant, interrupted exercise training during F/U, lost to F/U at our cardiologic clinic, a candidate of cardiac transplantation within 6 months, uncompensated HF patients, or a renal patient with an estimated glomerular filtration rate of < 30 ml/min/1.73m2, were not eligible for exercise training. Patients had absolute contraindications for cardiopulmonary exercise test (CPET) and aerobic activities, suggested by the American College of Sports Medicine (ACSM) [Pescatello, L.S.; Arena, R.; Riebe, D.; Thompson, P.D. ACSM’s guidelines for exercise testing and prescription, 9th ed.; Wolters Kluwer/Lippincott Williams & Wilkins: Philadelphia, PA., 2014], were also excluded from the study. Among 234 HF patients eligible for exercise trainings were further divided into exercise and non-exercise groups based on patient preference. All the participants provided informed consent before initiating the experimental procedure.)

3.     State also in subsection Clinical assessments, section Methods: "Age, sex,... and during F/U (RECORD item 12).

Reply: Thank you for your comments. The data in our study was collected directly from patients via questionnaires/experiments. We did not link these patients to other databases. However, we do feel that it would be much clearer for readers to know what variables were collected in this study. The list of all variables was provided in the supplemental table 1.

Reviewer 3 Report

The paper has improved after revision, but some issues remain.

My main concern is the high risk of selection bias. As patient preference was the determinant of group allocation it is not difficult to imagine that more severely diseased patients abstained from HIIT. Although propensity score matching may balance the groups, there is still a high risk of residual confounding. This must be stated as an important limitation of the present study. Further, because of this limitation the conclusion should be much more careful (e.g. HIIT might improve maximum exercise capacity and long-term survival in heart failure patients)

The low number of events during follow-up is a limitation and should be stated as such.

Regarding the Teichholz method for assessment of LVEF, the reference Lang et al. J Am Soc Echocardiogr 2005;18:1440 cannot be used to back up your statement “…it has been recommended to screen a large population”). In fact, Lang et al says: “The previously used Teichholz or Quinones methods of calculating LVEF from LV linear dimensions may result in inaccuracies as a result of the geometric assumptions required to convert a linear measurement to a 3D volume. Accordingly, the use of linear measurements to calculate LVEF is not recommended for clinic practice.” Instead of defending the Teichholz method in the present study, I suggest you acknowledge its use as a limitation.

Some editing of English language and style is still required.

Author Response

1.     My main concern is the high risk of selection bias. As patient preference was the determinant of group allocation it is not difficult to imagine that more severely diseased patients abstained from HIIT. Although propensity score matching may balance the groups, there is still a high risk of residual confounding. This must be stated as an important limitation of the present study. Further, because of this limitation the conclusion should be much more careful (e.g. HIIT might improve maximum exercise capacity and longterm survival in heart failure patients)

Reply: Thank you for your comment. Indeed, there may be other confounders affecting our interpretations. We have placed “Although propensity score matching may balance the groups, there is still a high risk of residual confounding” in Line 1764-1765, Limitations sub-section, Page 11/14. The conclusion for HIIT based on our observations has been modified as “This 8-year cohort study has demonstrated that HIIT improves the maximum exercise capacity of HF patients as previous investigations [Gossard et al. Am. J. Cardiol. 1986, 57, 446-449; Wisløff et al. Circulation 2007, 115, 3086-3094; Fu et al. Int. J. Cardiol. 2013, 167, 41-50; Wang et al. Int. J. Cardiol. 2013, 168, 1243-1250.; Ellingsen et al. Circulation 2017, 135, 839-84915; Fu et al. Am. J. Phys. Med. Rehabil. 2016, 95, 15-27], which may become a significant prognostic factor of long-term survival in HF patients” in the first sentence of the Conclusion section, Page 12/14.

2.     The low number of events during followup is a limitation and should be stated as such.

Reply: Thank you for your comment. We have described the small event number in Line 1765, Limitations sub-section, Page 11/14.

3.      Regarding the Teichholz method for assessment of LVEF, the reference Lang et al. J Am Soc Echocardiogr 2005;18:1440 cannot be used to back up your statement “…it has been recommended to screen a large population”). In fact, Lang et al says: “The previously used Teichholz or Quinones methods of calculating LVEF from LV linear dimensions may result in inaccuracies as a result of the geometric assumptions required to convert a linear measurement to a 3D volume. Accordingly, the use of linear measurements to calculate LVEF is not recommended for clinic practice.” Instead of defending the Teichholz method in the present study, I suggest you acknowledge its use as a limitation. Some editing of English language and style is still required.

Reply: Thank you for your comment. We have placed the disadvantage in Line 1766, Limitations, Page 11/14.

4.     Some editing of English language and style is still required.

Reply: Thank you for your suggestion. English editing has been performed again in our revised manuscript.